# Cardio PyMEA: A user-friendly, open-source Python application for cardiomyocyte microelectrode array analysis

**Christopher S. Dunham**[1]*, **Madelynn E. Mackenzie**[2], **Haruko Nakano**[3], **Alexis R. Kim**[1], **Atsushi Nakano**[3,4,5,6,7], **Adam Z. Stieg**[8,9]*, **James K. Gimzewski**[1,8,9]*

**1** Department of Chemistry and Biochemistry, University of California, Los Angeles, California, United States of America, **2** Department of Microbiology, Immunology & Molecular Genetics, University of California, Los Angeles, California, United States of America, **3** Department of Molecular, Cell and Developmental Biology, University of California, Los Angeles, California, United States of America, **4** Molecular Biology Institute, University of California, Los Angeles, California, United States of America, **5** Eli and Edythe Broad Center of Regenerative Medicine and Stem Cell Research, University of California, Los Angeles, California, United States of America, **6** Division of Cardiology, Department of Medicine, University of California, Los Angeles, California, United States of America, **7** Department of Cell Physiology, The Jikei University, Tokyo, Japan, **8** California NanoSystems Institute, University of California, Los Angeles, California, United States of America, **9** International Center for Materials Nanoarchitectonics (MANA), National Institute of Materials Science, Tsukuba, Japan

* csdunham@chem.ucla.edu (CSD); stieg@cnsi.ucla.edu (AZS); gimzewski@cnsi.ucla.edu (JKG)

**Data Availability Statement:** Software will be available through Github at the link provided in the paper, which is also provided here for convenience: https://github.com/csdunhamUC/cardio_pymea All

## Abstract

Open source analytical software for the analysis of electrophysiological cardiomyocyte data offers a variety of new functionalities to complement closed-source, proprietary tools. Here, we present the Cardio PyMEA application, a free, modifiable, and open source program for the analysis of microelectrode array (MEA) data obtained from cardiomyocyte cultures. Major software capabilities include: beat detection; pacemaker origin estimation; beat amplitude and interval; local activation time, upstroke velocity, and conduction velocity; analysis of cardiomyocyte property-distance relationships; and robust power law analysis of pacemaker spatiotemporal instability. Cardio PyMEA was written entirely in Python 3 to provide an accessible, integrated workflow that possesses a user-friendly graphical user interface (GUI) written in PyQt5 to allow for performant, cross-platform utilization. This application makes use of object-oriented programming (OOP) principles to facilitate the relatively straightforward incorporation of custom functionalities, e.g. power law analysis, that suit the needs of the user. Cardio PyMEA is available as an open source application under the terms of the GNU General Public License (GPL). The source code for Cardio PyMEA can be downloaded from Github at the following repository: https://github.com/csdunhamUC/cardio_pymea.

## Introduction

Cardiomyocyte cell cultures, particularly human embryonic and induced pluripotent stem cell-derived cardiomyocytes (hESC-CM and hiPSC-CM, respectively), are of significant

other data are to be provided through DataDryad and/or Zenodo at the following DOIs: Data: https://doi.org/10.5068/D14H5C Software Snapshot: https://doi.org/10.5281/zenodo.6462799 Software Supplemental (e.g. PIP requirements files): https://doi.org/10.5281/zenodo.6522426.

**Funding:** This work was funded by grants from the National Institutes of Health (https://www.nih.gov/grants-funding). Grant R21HL124503 was awarded to AZS, JKG, and AN. Grants R01 HL142801 and HL146159 were awarded to AN. The funders had no role in study design, data collection and analysis, decision to publish, or preparation of the manuscript.

**Competing interests:** The authors have declared that no competing interests exist.

interest for their potential to serve as model systems for studying a wide range of phenomena [1, 2]. Potential applications of hESC-CMs and hiPSC-CMs in scientific and industrial environments include: pharmacological drug screening, disease modeling, cardiac development and maturation, and regenerative medicine [3–11]. However, the utility of these cultures is often limited by their immature nature [12]. Currently, insufficient knowledge exists to enable these cells to fully achieve adult phenotype maturity *in vitro* [8, 10, 13]. This maturity deficiency limits our capacity to study disease states, including cardiomyopathy and late myocardial dysfunction, which may be linked to defective cardiomyocyte development in humans [9, 14–16]. The inability to induce maturation of stem cell-derived cardiomyocytes beyond the late fetal stage imposes limitations to their applications to disease modeling, drug screening, and regenerative medicine initiatives [4, 7–10].

In recent years, microelectrode arrays (MEAs) became a popular tool for use in the study of cardiomyocyte cultures [13, 17, 18]. Microelectrode arrays enable spatiotemporal analysis of cardiomyocyte field potentials, which are extracellular electrical potentials generated by the cells in the culture. Microelectrode arrays also confer the ability to perform controlled stimulation and pacing experiments of cardiomyocyte cultures using a variety of input waveforms [19, 20]. Concurrent with the increase in MEA-oriented cardiomyocyte analysis was the academic development of purpose-built MEA software with graphical user interfaces (GUIs) [17, 18, 21, 22]. These software applications enabled a wider audience to analyze electrophysiological cardiomyocyte data with open-access tools offering enhanced functionality beyond that offered by proprietary software released by the MEA system manufacturers. Each program offers a unique feature that provides value to electrophysiological cardiomyocyte analysis, e.g. new methods for determining the T-wave endpoints and for calculating conduction velocity.

Although powerful and incredibly helpful for cardiomyocyte analysis, there is one significant hurdle that merits consideration. Most, if not all, previously released open source cardiomyocyte analysis tools were written in MATLAB, a licensed programming language developed by MathWorks (Natick, MA) for engineers and scientists [23]. Despite its relative maturity, MATLAB's licensing can impose both financial barriers and accessibility constraints on continued application development. Such costs and constraints can pose significant challenges to research communities [24]. The Python programming language overcomes these hindrances thanks to its free and open source nature. Thus, the financial barrier inherent to developing MATLAB-based cardiomyocyte tools can be overcome by developing Python-based tools instead. This perspective motivated the development of a new MEA analysis application: Cardio PyMEA.

Cardio PyMEA is a free and open source software application (FOSS) written in Python for the analysis of MEA recordings of cardiomyocyte cell cultures. It was designed with a user-friendly GUI to allow scientific programmers and non-programmers alike to engage in robust electrophysiological analysis of MEA data. Because Cardio PyMEA allows the end user to readily adjust analysis parameters, the software is capable of analyzing noisy data sets that demonstrate incompatibility with automated algorithms and applications. In addition to calculating common cardiomyocyte parameters (e.g. pacemaker origin, local activation time, conduction velocity, etc.), Cardio PyMEA offers unique features such as property-distance relationship analysis and power law analysis. The latter is particularly useful for its applications in understanding cardiomyocyte culture system dynamics [25].

Unlike most commercial systems, which are fixed in capabilities, Cardio PyMEA was developed to facilitate relatively easy extensibility thanks to its utilization of object oriented programming principles. This makes the addition of new GUI elements (e.g. additional plotting windows) and new calculations (e.g. machine learning techniques) comparatively straightforward to achieve. Here, Cardio PyMEA is presented using real data acquired from MEA

recordings of hESC-CM and hIPSC-CM cell cultures to demonstrate its utility in cardiomyocyte characterization and analysis, and its potential for continued development.

## Materials and methods

### Cell cultures and microelectrode array measurements

Human ESCs were grown and differentiated in a chemically defined condition as previously described [13, 26, 27]. Usage of all human embryonic stem cell lines is approved by the UCLA Embryonic Stem Cell Research Oversight (ESCRO) Committee and the Institutional Review Boards (IRB) (approval #2009-006-04). After differentiation, cardiomyocytes were plated as two-dimensional monolayers on matrigel-coated (Corning #354277), commercially available microelectrode arrays (MEAs) containing 120 integrated TiN electrodes. These electrodes were 30 μm in diameter and were manufactured with an interelectrode spacing of 200 μm (Multichannel Systems, Reutlingen, Germany). Following plating, the MEAs were placed in an incubator set to 37˚C with a gas flow of 5% $CO_2$. The cell cultures were given no less than 24 hours to ensure the cardiomyocytes adhered well to each MEA.

### Software requirements

Cardio PyMEA is written for Python 3.8 or above and utilizes several freely available, actively maintained Python libraries. These libraries include NumPy, SciPy, Pandas, Matplotlib, Seaborn, and Numba, among others. The GUI was constructed using PyQt5. The complete source code and the dependency (pip) requirements file ("requirements.txt") for Cardio PyMEA are available on Github at the following link: https://github.com/csdunhamUC/cardio_pymea.

### Using Cardio PyMEA

The preferred way to launch Cardio PyMEA is through the terminal of your chosen operating system. The central file used to run and operate Cardio PyMEA is 'analysisGUI.py'. This file contains the code for the various graphical windows, import function, MEA dictionary, and other functions. Alternatively, for users less inclined toward operating Cardio PyMEA from a terminal, executable files (made using PyInstaller) for Linux, MacOS, and Windows 10 can be accessed from the DataDryad repository.

   The main window of Cardio PyMEA, shown in Fig 1, consists of a traditional menu navigation system anchored around the central plotting window that, after calculations, will display heatmaps of the time lag (pacemaker), local activation time (LAT), upstroke velocity (dV/dt), and conduction velocity (CV). To import MEA data, the user can click on 'File' → 'Import Data' and select an appropriate MC_Data-derived *.txt file. The expected format of MEA data is discussed in the next section. Once the data is imported, the user can run the beat detection by selecting 'Calculations' → 'Find Beats', choosing whether to apply any smoothing filters, and then clicking the button to execute. After this step, the user can choose among the various calculations of interest, including 'All' to perform the calculations for pacemaker, LAT, maximal upstroke velocity ($dV/dt_{max}$), CV, beat amplitude, and beat interval properties. This workflow is summarized by the flowchart in Fig 2.

### Input data format and geometric configuration of MEAs

Cardio PyMEA was designed to utilize data obtained from Multichannel Systems' MC_Rack MEA recording software and the MC_Data conversion tool. The *.mcd files produced by MC_Rack must be converted to ASCII (*.txt) files using MC_Data. Cardio PyMEA expects that the resulting *.txt file should be organized column-wise, beginning with the time, t, in

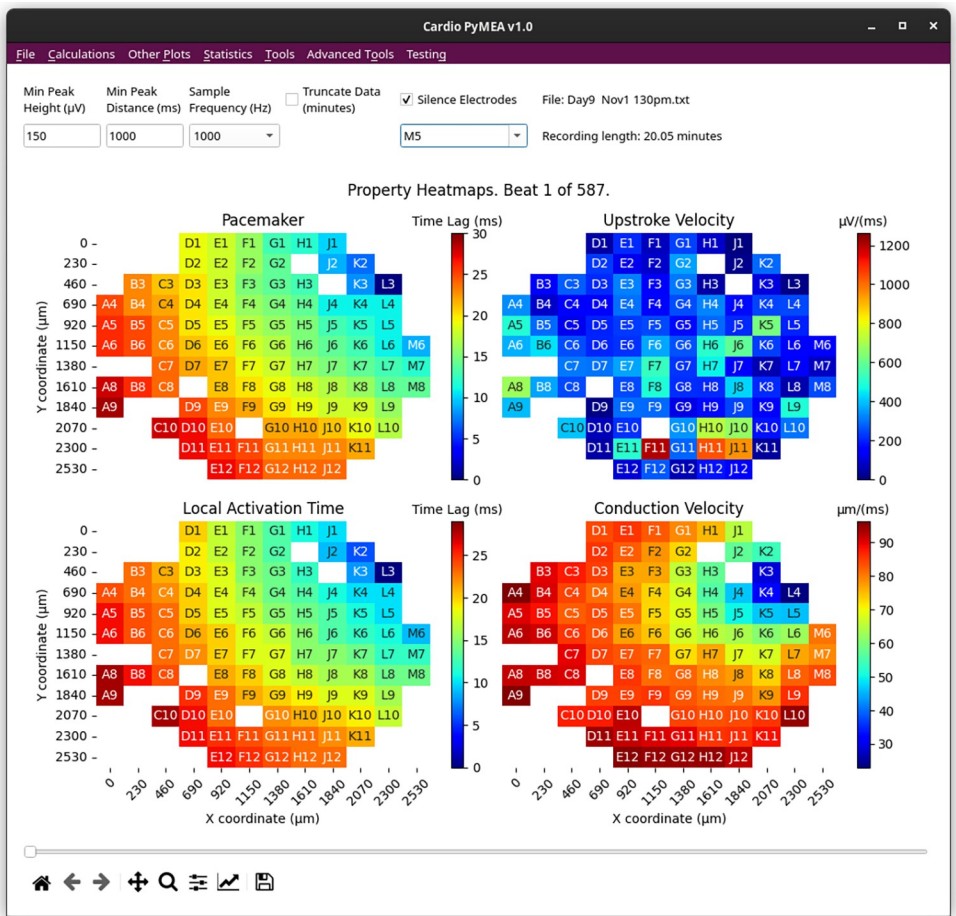

**Fig 1. Main window of Cardio PyMEA graphical user interface.** All of the user's interactions with Cardio PyMEA stem from this window. The user can utilize the various drop-down menu options to import data, perform calculations, and perform statistical analysis.

milliseconds (or fractions of milliseconds for sample frequencies > 1 KHz), followed by electrode names with their measured voltages (in microvolts, μV). For example, when using a 120 electrode MEA recording with a sample frequency of 1000 Hz, the first column of the *.txt data file is expected to correspond to the time, t, in milliseconds. The second column is expected to correspond to the electrode F7. This electrode is designated as channel 1 (electrode 1) in the manufacturer's MEA channel schematic. Each subsequent electrode is expected to adhere to this schematic. As a result, Cardio PyMEA expects that the *.mcd to *.txt conversion proceeds with 'All' electrodes selected in the conversion wizard window.

Cardio PyMEA's design allows for the relatively easy expansion of other MEA geometries due to its object-oriented nature. The ElectrodeConfig class (a template for creating an object; in this case, an electrode configuration variable) houses information for the MEA configurations that a researcher may choose to employ. Currently, Cardio PyMEA recognizes both 60 and 120 electrode MEAs from Multichannel Systems with 30 μm diameters and 200 μm interelectrode spacings. Other configurations can be added in a straightforward manner by creating new class attributes (a variable that belongs to the class) for the other systems in the ElectrodeConfig class. Additionally, a tutorial for supplementing Cardio PyMEA with additional MEA geometries is provided in S1 File.

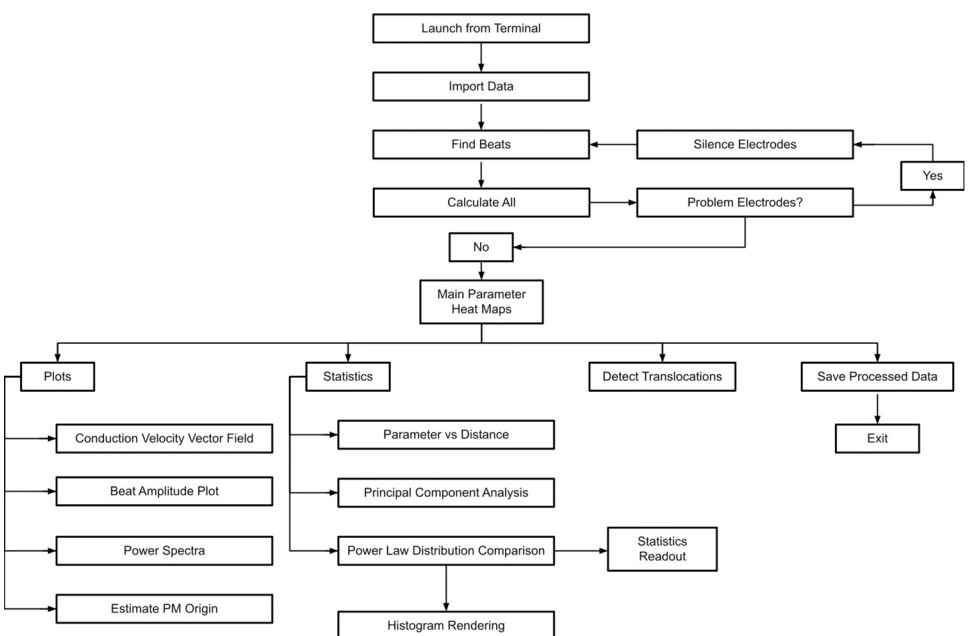

**Fig 2. Operational flowchart for using Cardio PyMEA.** This flowchart illustrates how Cardio PyMEA can be operated in order to analyze cardiomyocyte MEA data.

All MEA data used in Cardio PyMEA requires geometric coordinates and labeling for the electrodes in the array. The schematic used in Cardio PyMEA arbitrarily positions the coordinate map origin (i.e. the coordinate (0, 0)) in the top-left corner of the array. A schematic illustrating the expected geometries is shown in Fig 3. Electrode coordinates are assigned manually to each electrode and are housed in a Python dictionary for each respective MEA configuration (i.e. each ElectrodeConfig attribute).

## Cardio PyMEA calculation methods and functions

One of the strengths of Cardio PyMEA is that it houses a wide variety of calculations used in the analysis of cardiomyocyte MEA data. These functions and calculations include: automated electrode exclusion algorithm, manual electrode silencing, pacemaker, LAT, dV/dt$_{max}$, CV, field potential duration (FPD), beat amplitude, beat interval, and pacemaker translocations (instances in which the spatial configuration of the pacemaker region becomes unstable and translocates, or moves, to another area of the MEA over time) [25]. A complete description and representative example of each calculation output follows.

**Beat detection.** Cardiomyocyte beats, identified in MEAs as the most prominent peak of the field potential, are calculated by Cardio PyMEA using the findpeaks function contained in the SciPy library. First, users enter their designed peak height (minimum signal amplitude, in microvolts) and peak distance (minimum separation between peaks, in milliseconds) into the appropriate fields in the GUI. Alternatively, users may proceed using the default parameters. Next, beat detection can be performed by selecting 'Calculations' from Cardio PyMEA's menu bar and choosing 'Find Beats'. This action opens the 'Find Beats Results' window, shown in Fig 4. The user can next choose whether to filter the signal using low-pass, high-pass, or band-pass Butterworth filters, or use the raw (unfiltered) signal. Finally, the user clicks the 'Find Beats' button, which begins the calculation using the given parameters. Two plots are subsequently generated to show the MEA's signal trace of voltage (μV) vs time (ms). The left-side

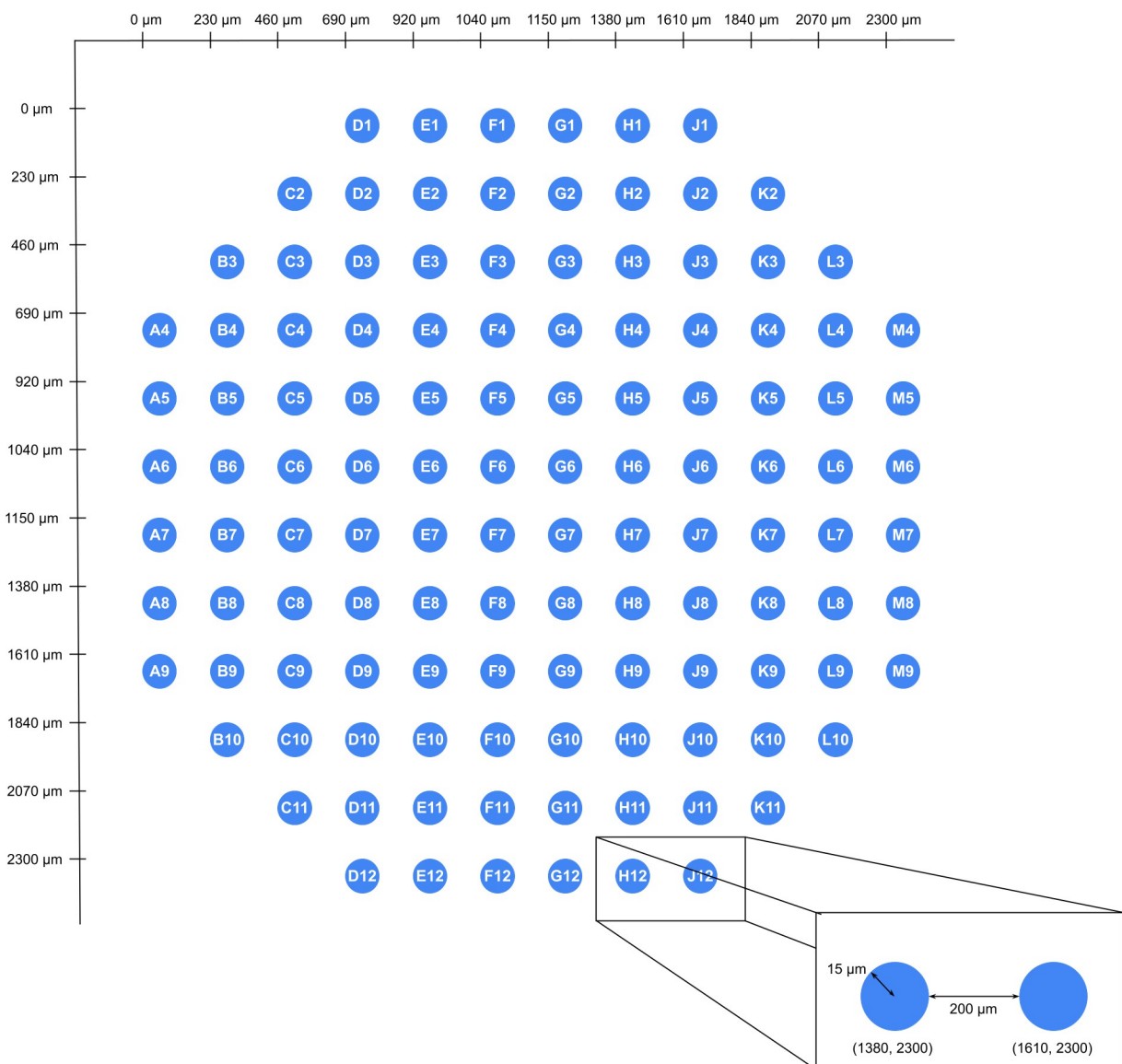

**Fig 3. Representative schematic of the geometric configuration used for a 120 electrode MEA.** This coordinate system shown here was devised using an interelectrode spacing of 200 μm and electrode diameter of 30 μm.

plot shows a single beat for a single electrode. This view can be adjusted by using sliders marked with 'Beat' and 'Electrode', respectively. The right-side plot shows a condensed view of field potentials for all electrodes at the selected beat. These plots can be further manipulated using the provided navigation toolbar. Finally, these plots can be saved as *.png files by clicking the 'Save' (disk) icon in the navigation toolbar.

**Automated electrode exclusion.** Electrodes with an atypical beat count, as measured in comparison to other electrodes, may negatively affect the calculations performed by Cardio PyMEA (e.g. by producing vectors and matrices of different dimensions, requiring interpolation or other data pre-processing to resolve). To rectify this, Cardio PyMEA utilizes an automated electrode exclusion process. After determining the number of beats in a data set using the beat detection algorithm described previously, Cardio PyMEA calculates the beat count

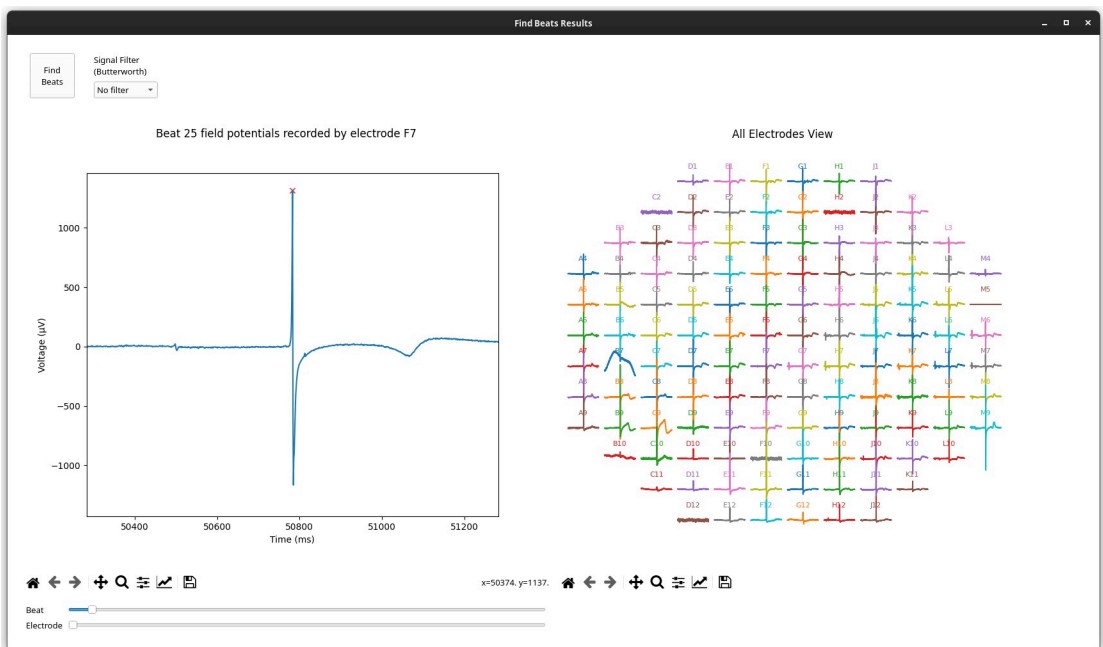

**Fig 4. 'Find Beats Results' window.** The left plot shows an individual beat for the electrode specified by the window slider. The sliders below the plot control which beat and/or electrode is plotted. The right plot shows the full MEA for a specified beat and is controlled by the same 'Beat' slider as the left plot.

mode across the data set for each electrode. Following this calculation, Cardio PyMEA automatically excludes those electrodes whose beat counts vary from the beat count mode.

**Manual electrode silencing.** It is possible for electrodes to pass the automated electrode exclusion process yet still exhibit unusual signal characteristics, e.g. inverted field potential peaks. These problematic electrodes can be seen qualitatively in either the 'Find Beats Results' plots or in property heatmaps, where they contradict the behavior of their neighbors during certain beats. Electrodes demonstrating atypical or erratic behavior (e.g. abnormally high time lag values) can be manually silenced by selecting the check-box labeled 'Silence Electrodes' in the top right corner of the main window. Once checked, the user can choose which electrode (s) to silence from the drop-down menu. After choosing which electrodes to silence, 'Find Beats' must be executed again to complete the exclusion. Once this operation is performed, data from the manually excluded electrodes will no longer be included for analysis. Electrode exclusion can be reversed by simply de-selecting the silenced electrode(s) and executing 'Find Beats' again.

**Time lag (pacemaker) calculation and origin estimation.** For each electrode in the MEA, Cardio PyMEA records the time at which the field potential peak of an individual beat is detected. Cardio PyMEA then compares these values to determine the point of origin of the pacemaker signal. The lowest time value (i.e. earliest electrode to detect a signal during a beat) is determined to correspond to the pacemaker region. To normalize the data, Cardio PyMEA subtracts the field potential peak time recorded across all other electrodes from the earliest time recorded at the pacemaker, as shown in Fig 5A. This means that the pacemaker electrode is the one with the lowest time lag value, and in the normalized data, corresponds to the electrode with a time lag value of 0 ms. This method is repeated for each beat and yields the time lag (pacemaker) value: the time elapsed between the earliest signal detected and the time signal is detected at another electrode in the array. Cardio PyMEA uses the normalized time lags to

render a color-coded heat map which is spatially defined by the geometry of the MEA. Electrodes colored dark blue indicate the location of the pacemaker (minimum time lag), while dark red indicates electrodes with the highest time lag farthest from the pacemaker (maximum time lag).

To access this operation, select 'Calculations' → 'Pacemaker' from the main window menu bar. A new window will open that displays the pacemaker heat map for the first beat detected in the data set. The x and y axes provide the 2D coordinates of each electrode. The slider at the bottom of the window is used to toggle between heat maps for all beats detected in the given data set.

The coordinates of the pacemaker origin can also be estimated by Cardio PyMEA. First, the estimation employs a contour plot to determine the wave front of the pacemaker data. The process is designed to favor the contour line that contains the most data points for model fitting. Next, Cardio PyMEA uses nonlinear least squares to estimate the coordinates of the center (h, k) and the radius (r) for a circle defined by the equation

$$r = \sqrt{(x - h)^2 + (y - k)^2}$$

The radius parameter, r, is bound within the geometry of the MEA during the fitting process. This measure is taken to ensure that the center of the circle, (h, k), does not reside outside of the culture region of the MEA. Ultimately, the calculated coordinate (h, k) represents the estimated location of the pacemaker origin.

To access this operation, select 'Other Plots' → 'Estimated Pacemaker Origin' from the main window menu bar. A new window will open that depicts a contour map. This contour map indicates the estimated origin of the pacemaker for a single beat. The slider at the bottom of the window can be used to change which beat is displayed. The orange dot at the center of the rendered circle denotes the estimated origin of the pacemaker signal, while the blue dots at the edge of the circle represent the values from the wave front that were used during fitting.

**Local activation time calculation.**   The LAT is calculated by finding the maximal negative intrinsicoid deflection (the maximum negative derivative (-dV/dt) to the right of the field potential peak in a given beat). For every electrode, Cardio PyMEA calculates the derivative (slope) of the field potential signal, beginning from the peak and moving to the right, for each beat in the data set, as shown in Fig 5B. The time (x) associated with the maximum negative derivative represents the LAT. These calculated LAT values are normalized in a similar manner to the time lag values for the pacemaker calculations. Using the calculated LATs, Cardio PyMEA renders a heatmap for qualitative data analysis.

To access this operation, select 'Calculations' → 'Local Activation Time' from the main window menu bar. A new window will open that displays the LAT heatmap. This heat map uses a color scheme that is identical to the pacemaker heat map: dark blue indicates the minimum, normalized LAT, while dark red indicates the maximum, normalized LAT in the full recording. The x and y coordinates represent the position of the specified electrode in the MEA system. The slider at the bottom of the window can be used to change which beat is displayed.

**Maximum upstroke velocity calculation.**   The maximum upstroke velocity (dV/dt$_{max}$) refers to the maximum slope to the left of the field potential peak, as shown in Fig 5B. Cardio PyMEA calculates the derivative, using a backward finite difference method, of the time series data to the left of the field potential peak (i.e. preceding the peak) for each electrode and each beat. Finally, Cardio PyMEA normalizes the heatmap color gradient to the observed global minimum and maximum dV/dt$_{max}$ values.

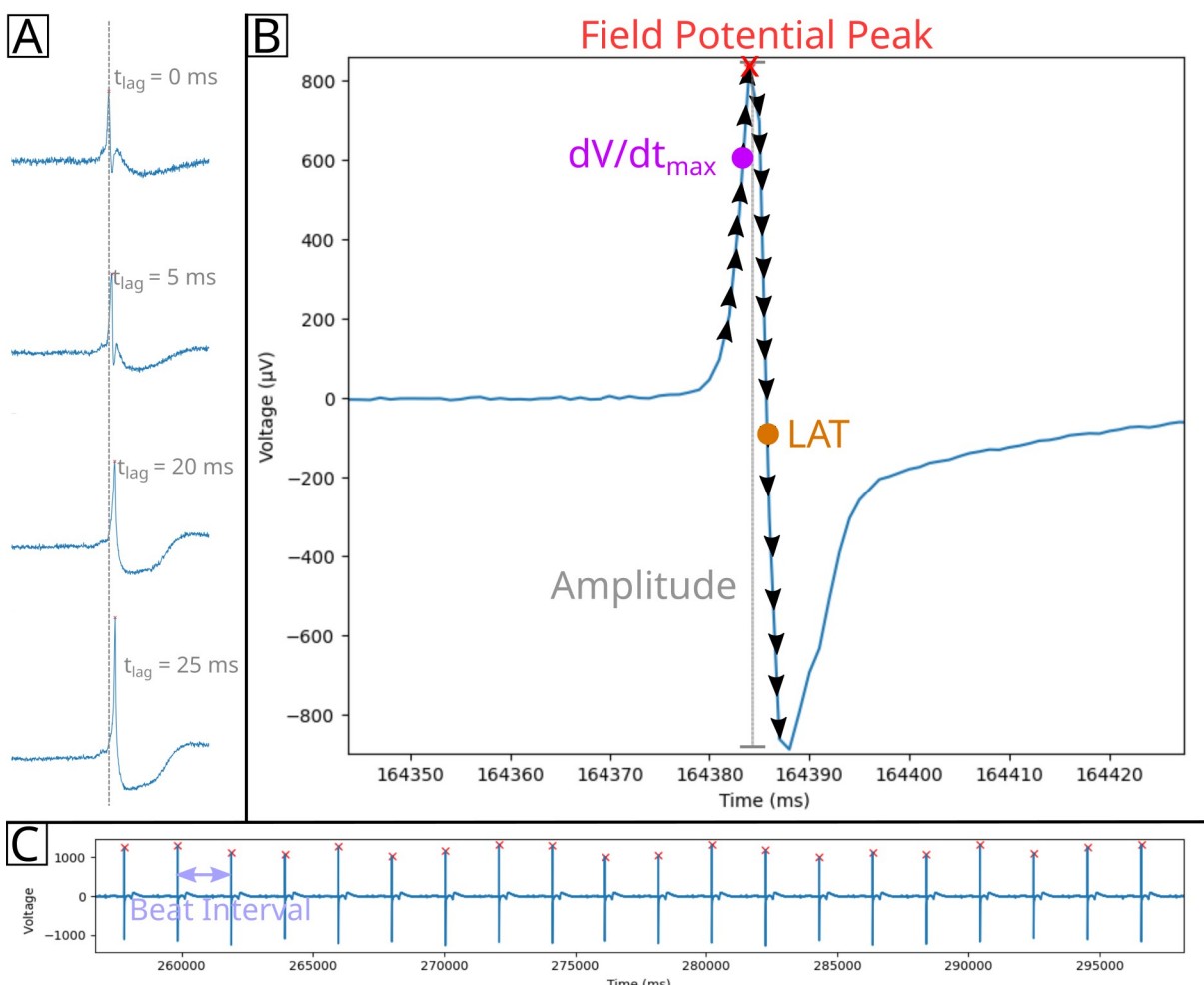

**Fig 5. Diagram of cardiomyocyte field potential property calculations.** A) Time lag ($t_{lag}$, pacemaker) calculation. The field potential peak for each electrode is calculated, then the peak with the minimum time is normalized to 0 ms. This process is applied to all electrodes to yield normalized time lag values for the given beat. B) Amplitude (gray vertical line), $dV/dt_{max}$, and LAT can be calculated from the field potential signal. Black arrows indicate derivative calculations along points prior to or after the field potential peak (red x). The maximum positive ($dV/dt_{max}$, magenta circle) and negative (LAT, orange circle) derivatives are identified and used in calculating $dV/dt_{max}$ and LAT, respectively. C) Beat intervals are calculated by determining the time between field potential peaks.

To access this operation, select 'Calculations' → 'Upstroke Velocity' from the main window menu bar. A new window will open that displays a heat map for the $dV/dt_{max}$ data of each electrode for the specified beat. The legend to the right describes the color coding: dark blue corresponds to the observed minimum, while dark red corresponds to the observed maximum. The slider along the bottom enables users to toggle between data for each individual beat.

**Beat amplitude and beat interval calculations.** Beat amplitude refers to the voltage magnitude of the primary peak (R-wave-like peak) of the field potential for a detected beat. Because these potentials encompass both positive and negative peaks, the beat amplitude calculation in Cardio PyMEA identifies both the positive and negative field potential signals at the location of the R-wave-like peak when determining the overall magnitude of the beat [19]. This is indicated in Fig 5B. The beat interval is calculated by identifying the time that Beat A occurred and subtracting that value from the time that Beat B occurred. This difference reflects the interval between the beats and is shown in Fig 5C. Beat intervals are calculated this way for all beats in

the recording. Cardio PyMEA uses data from the LAT calculation to compute these differences and generate beat intervals.

**Conduction velocity calculation.** Conduction velocity is defined as the speed and direction of the propagation of an electrochemical signal or impulse along a pathway in a network of cells. Conduction velocity is calculated in a straightforward manner by determining the distance between any two electrodes and dividing this distance by the change in local activation time ($\Delta$LAT) recorded at the electrodes, i.e.: (electrode distance)/$\Delta$LAT. This is Cardio PyMEA's default CV calculation method for 2 reasons: 1) it is a simple, computationally efficient calculation and 2) the precise geometry of the MEAs provides a uniform map that ensures consistent, reasonable results across cultures. This method is similar to one used in a previous study [13].

**Field potential duration calculation.** Cardio PyMEA uses previously described methods to calculate the FPD [28]. Here, the R-wave is analogous to the field potential peak and the T-wave endpoint is determined using the method from Vázquez-Seisdedos et al. [28]. A brief summary of this method follows. First, the T-wave is identified using a peak finding algorithm akin to the R-wave. Second, the maximum first derivative of the field potential signal between the T-wave peak and baseline is identified. Using the location (i.e. time or x-value) of the maximum derivative as a fixed point, along with an arbitrary location 50–200 ms past the T-wave peak, the algorithm fits a trapezoid with a mobile point to the two fixed points. Optimizing the maximum area of the trapezoid yields the T-wave endpoint [28]. The length of time (in milliseconds) between the R-wave and T-wave endpoint constitutes the FPD. In addition to calculating the FPD, Cardio PyMEA will generate a plot that can be cycled through each beat and each electrode using the slider at the bottom of the window. This window also facilitates visual confirmation of the T-wave endpoint in the user's data.

**Pacemaker translocation detection algorithm.** Pacemaker translocations were identified through monitoring the movement of the pacemaker region across subsequent beats. Provided that this movement (translocation) exceeds a distance threshold (500 µm), a timer is engaged to count the number of beats, along with the time interval, that the pacemaker region is stable in the new location. This period of stability is referred to as a "quiescent period" (i.e. the pacemaker is stable, or tranquil, and does not move during this time). If the location of the pacemaker region changes again in a manner that exceeds the distance threshold, the timer is stopped, the quiescent period (in beats and recording time) is stored in a list, and the timer is reset for the new position. The process repeats for each detected pacemaker region of each beat over the full MEA recording. Once the calculation concludes, the first recorded quiescent period is excluded from the list. This is done to remove potential artifacts induced by the uncertainty surrounding the true duration of the quiescent period of the pacemaker region. Similarly, the end of the recording does not contribute to a translocation ("event") designation or quiescent period. The algorithm was verified manually through visual inspection and computationally in Python in order to ensure agreement of results.

## Statistical analyses unique to Cardio PyMEA

Statistical analysis of cardiomyocyte culture data can take a variety of forms. Cardio PyMEA output data can be saved as a spreadsheet file (*.xlsx format) via the save data function (described in an upcoming section) for further analysis by the user. Additionally, Cardio PyMEA provides two types of statistical assessment that are unique to the software in the realm of cardiomyocyte analysis: 1) analysis of property-distance relationships and 2) analysis of power law behavior in pacemaker translocations.

**Property-distance relationship analysis.**   Property-distance relationships (also referred to here as 'Property vs Distance') can be evaluated to assess whether pacemaker time lag, LAT, upstroke velocity, and/or CV exhibit correlations with the distance from the estimated pacemaker origin. The user can perform a simple elimination of outliers by designating a number of standard deviations ('Sigma') from the mean. Any values outside of the range specified by the (number of standard deviations $*$ Sigma) operation are excluded from the analysis.

To access this operation, select 'Statistics' $\rightarrow$ 'Property vs Distance'. Using this function, plots for each property versus distance from the pacemaker region are produced. Either linear (pacemaker, LAT, $dV/dt_{max}$) or nonlinear (CV) regression is applied to obtain a goodness-of-fit metric, $R^2$. The top 10 $R^2$ values for each property, and their associated beats, are shown in a text box on the right side of the window.

**Power law analysis and distribution comparisons.**   Power laws are heavy-tailed probability distributions of the form $P(X) \propto x^{-\alpha}$. Demonstration of power law relationships between measurable properties in the system of interest, along with the calculated value of the power law exponent, , can provide unique information regarding the underlying properties of a complex system. These properties may include scale-free dynamics, fractal geometries, and long-range spatiotemporal correlations, among others [29–31]. The analysis of power laws is particularly important in the study of cardiac systems because power law exponents could also provide diagnostic value. Previous studies in patients with myocardial infarction, coronary heart disease, and heart transplants showed that the value of  measured for heart rate variability (small variations in the beat interval between heart beats) differed significantly from healthy patients [32–36]. Therefore, analysis of power laws in cardiomyocyte cultures could provide unique and invaluable insight into the underlying culture conditions and system dynamics.

Power law analysis in Cardio PyMEA relies upon several functions contained within the powerlaw Python library [37]. Cardio PyMEA uses the distribution_compare method from powerlaw (with the normalized_ratio parameter set to True) to compare how well a given data set adheres to other, similarly heavy-tailed, probability distributions [37]. This method calculates the log-likelihood ratios (LLRs) and p-values in order to compare distributions and determine the significance of the results. If the LLR is positive, then the first distribution (power law) is the best fit; if the LLR is negative, then the second distribution tested is a better fit. This allows users to determine if the data robustly demonstrate power law behavior, or if another, similarly heavy-tailed distribution serves as a superior descriptor of the data.

Cardio PyMEA also provides a qualitative assessment of the data, rendering a histogram superimposed with probability distribution curves fit to the entered data (default: power law, log-normal, stretched exponential). The user can specify the number of bins used in the histogram or rely upon automated methods (e.g. Sturges' Rule) to determine the number of bins for them [38]. To access these functions, users first must detect any translocations in the data set using 'Tools' $\rightarrow$ 'Detect Translocations' and then select 'Statistics' $\rightarrow$ 'Power Law Distribution Comparison' from the menu bar. Subsequently, a new window will open that displays qualitative plots and quantitative evaluations for power law, exponential, log-normal, Weibull, and doubly truncated power law probability distributions.

## Saving data

Cardio PyMEA allows the user to save their processed data for each calculated parameter set. These data include values for: pacemaker, LAT, $dV/dt_{max}$, CV, beat amplitude, beat interval, and property vs distance statistics. The save function writes the data to a single, multi-tab spreadsheet ($*$.xlsx) file. The file can be opened in FOSS spreadsheet software, e.g. LibreOffice Calc, or through licensed software, e.g. Microsoft Excel. To access this operation, select 'File'

→ 'Save Processed Data' from the main window's menu bar. The file name can be inputted and saved to the user's chosen directory. An example save file is provided in S2 File.

## Batch analysis

Cardio PyMEA affords batch processing of pacemaker translocation data to simplify power law analysis for large numbers of MEA recordings. The user simply needs to add their file information to the batch file spreadsheet (included in the repository). Once the batch file is ready, the user will navigate to 'File' → 'Batch' and select their spreadsheet (*.xlsx) batch file. At that point, Cardio PyMEA will detect and compile pacemaker translocations for all files in the batch. Power law analysis can then be performed using the batched data, which Cardio PyMEA will recognize and require no additional work on the part of the user. An example batch file is also provided in S3 File.

## Results and discussion

### Cardio PyMEA grants users flexibility via parameter control during cardiomyocyte analysis

Cardio PyMEA was tested using 30 MEA recordings across 3 distinct differentiations. Cardiomyocytes demonstrated good adhesion to the MEAs (Fig 6A and 6B). The beat detection algorithm was successful in identifying beats from the field potential signals without the need for signal filtering (Fig 6C). The beat detection algorithm was validated manually for each MEA and performed well under varying levels of noise, as shown in Fig 6D–6F. Faulty electrodes,

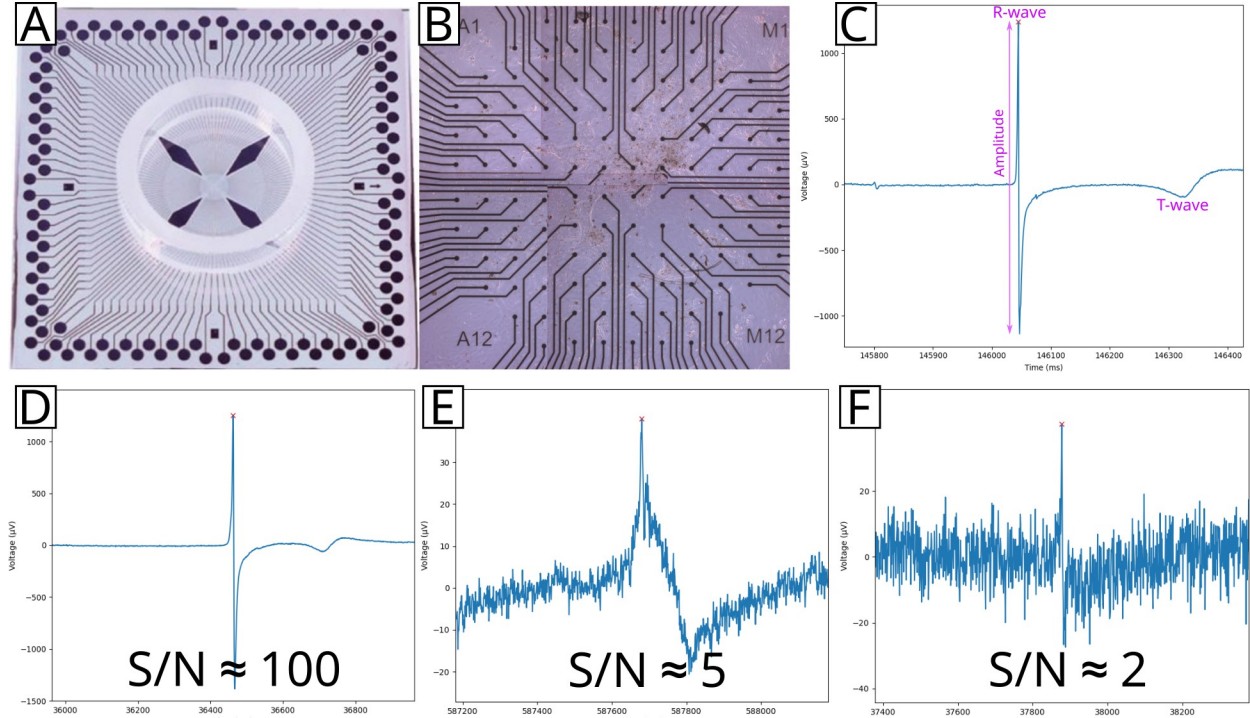

**Fig 6. Beat detection results for an MEA-plated cardiomyocyte culture.** An individual 120 electrode MEA (A) was plated with cardiomyocytes (B). Field potentials were recorded (C) and analyzed to determine the R-wave-like peak, beat amplitude, T-wave endpoint, and other features. These field potentials showed varying levels of noise, ranging from clean (D) to moderately (E) or significantly (F) noisy, as indicated by the decreasing signal-to-noise (S/N) ratios. Beat detection was performed for all field potentials across all MEAs.

whether due to cell culture conditions (e.g. non-uniform spread of the culture), heterogeneous Matrigel application causing depressed local signal detection, or electrode degradation as a result of prolonged MEA usage, were excluded by the automated electrode exclusion algorithm. The excluded electrodes were deemed appropriate upon manual review, thus validating the algorithmic choices. The remaining electrodes demonstrated sufficient signal for all subsequent analyses. Successful cultures typically had only a few electrodes excluded from the analysis. The average beat rate (beats per minute, bpm) across all datasets was 36.64 bpm with a standard deviation of 14.48 bpm.

Cardio PyMEA demonstrated effective performance under noisy conditions thanks to its utilization of user-provided parameter inputs. User-defined thresholds of beat amplitude and distance provide flexibility in analyzing noisy datasets, even under low signal-to-noise (S/N) conditions (Fig 6E and 6F). Successful beat detection enables the analysis of a variety of cardiomyocyte culture properties, including: pacemaker time lag, LAT, upstroke velocity, CV, FPD, and beat amplitude and interval. Local activation time, which is associated with the time of maximum sodium conductance in the myocardium, is a requisite property for analyzing cardiac conduction velocity and serves as another metric for assessing pacemaker activity. The maximum decrease in field potential voltage is generally accepted as the activation time for a beat [39]. Maximum upstroke velocity is associated with the peak influx of sodium and calcium ions during the action potential and presents another property useful for evaluating cardiomyocyte maturity [40]. Finally, beat amplitude and interval are widely known to serve as indicators of cardiomyocyte maturity.

## Cardio PyMEA provides effective calculation and estimation of pacemaker regions in 2D cardiomyocyte cultures

Cardio PyMEA successfully generated per-beat, spatial heatmaps of time lag (pacemaker) activity across the MEA culture, as shown in Fig 7A–7C. Pacemaker heatmaps were evaluated across dozens of beats and multiple MEAs. Subsequent analysis of the time lag wave front propagation using the pacemaker origin estimation tool revealed the most likely signal origin for the given data (Fig 7D–7F). Identification of the pacemaker region in the culture could have practical implications for MEA stimulation experiments, e.g. by providing information regarding which electrode(s) to use for stimulation of a spatially-constrained system [41]. These results demonstrate Cardio PyMEA's ability to produce spatially defined heatmaps of pacemaker activity and determine a logical point-of-origin for the active pacemaker region within the culture.

## Cardio PyMEA provides unique statistical tools for assessing cardiomyocyte maturation

During natural heart formation in the early embryo, all cardiomyocytes develop autonomic contraction and possess pacemaker characteristics. During the mid-gestational stages, working myocardium diversifies and acquires characteristic phenotypes including decreased automaticity, higher CV, and higher contractility [42]. If the stem cell-derived cardiogenesis recapitulates embryonic heart formation, one could hypothesize that cells distant from the pacemaking cells on an MEA present an increase in CV and field potential magnitude. To investigate this possible distance dependency of cardiomyocyte properties, Cardio PyMEA provides a unique function to analyze changes in property values with distance (i.e. property-distance relationships). Several plots were constructed tracking the correlations between a) pacemaker time lag, b) LAT, c) $dV/dt_{max}$, and d) CV and distance from the pacemaker region, both on a per-beat basis and averaged across all beats in the dataset. An example output is provided in Fig 8 and

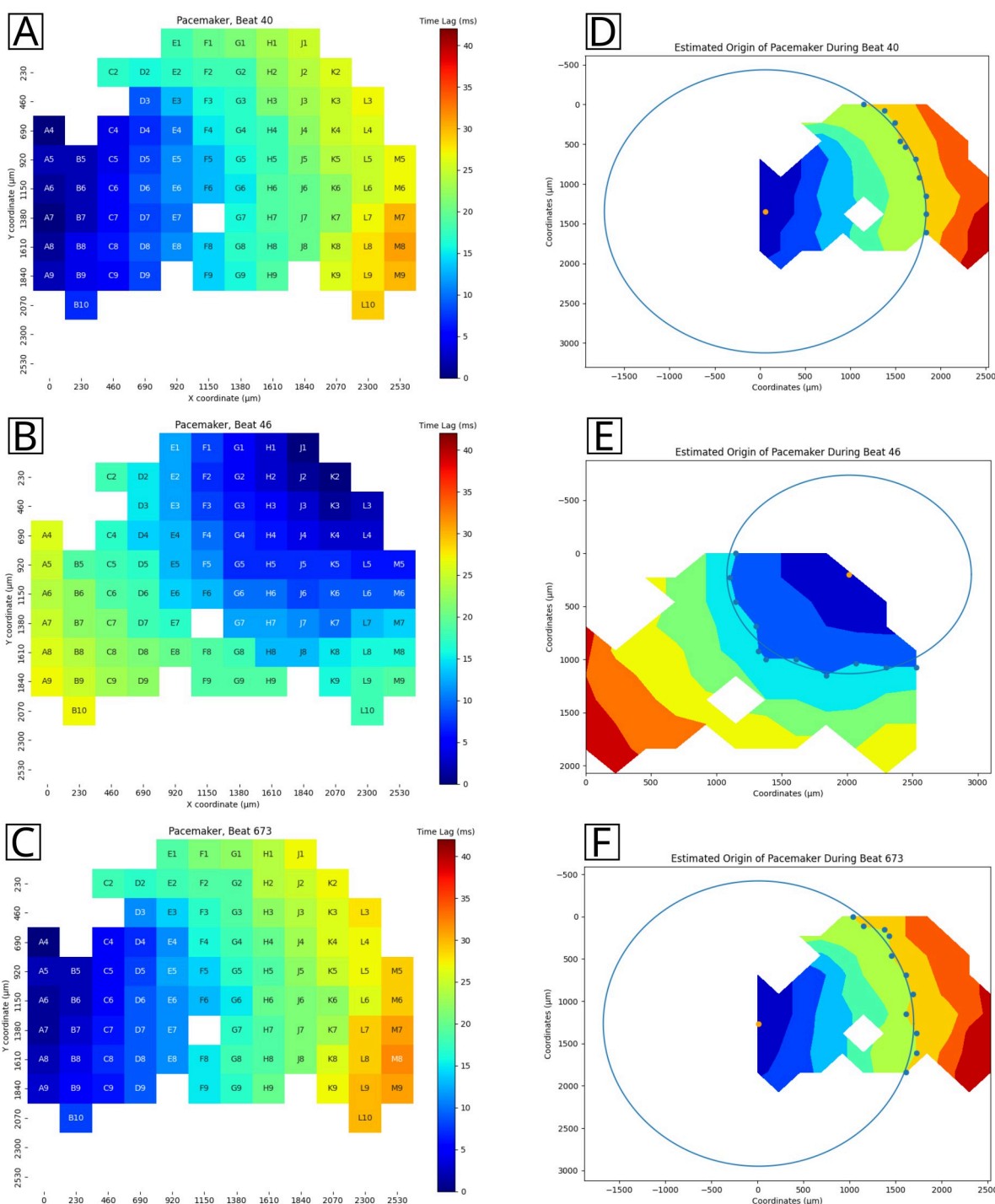

**Fig 7. Pacemaker analysis output.** A-C: representative pacemaker heatmap, spatially defined by the MEA coordinate system. D-F: estimated pacemaker location for the given data based on the time lag propagation wavefront. The estimated origin is indicated by an orange dot.

yields a mean $R^2 = 0.898$ for conduction velocity versus distance, suggesting that a relationship exists, based on good fitting between these two properties, for the given cardiomyocyte culture. These data suggest that the electrophysiological activity of *in vitro* stem cell-derived

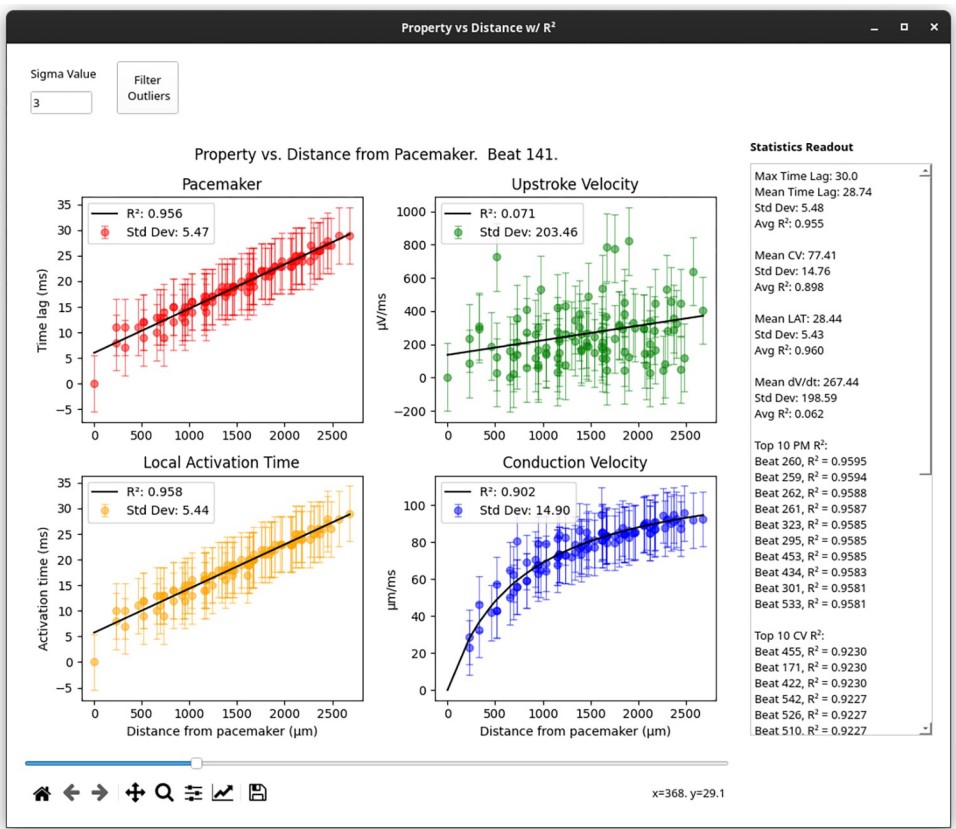

**Fig 8. 'Property vs Distance with R² window.** Representative plots of each property versus distance from the pacemaker. Statistical quantities, such as average time lag and conduction velocity, as well as top $R^2$ values, are presented in the 'Statistics Readout' text box on the right side of the window.

cardiomyocyte cultures may reflect early events during the lineage diversification and maturation of cardiac conduction system and working myocardium.

## Cardio PyMEA provides unique tools for and simplifies the analysis of pacemaker translocations for power law behavior

A unique feature provided by Cardio PyMEA is the ability to investigate the quiescent period between pacemaker translocations in two-dimensional cardiomyocyte cultures as a potential power law-obeying phenomenon. Here, Cardio PyMEA was used in combination with the powerlaw Python library to evaluate pacemaker translocations [25, 37]. The justifications for these comparisons were based on their conceptual similarity to neuronal avalanche inter-burst (also sometimes referred to as inter-event) intervals, or IBI [43, 44].

Cardio PyMEA simplifies power law analysis of cardiomyocyte cultures by effectively wrapping a graphical user interface over the powerlaw Python library, allowing the user to easily select from several input parameters [37]. Furthermore, the user can choose to evaluate power law behavior at both the individual recording level and in large batches (e.g. 30 or more) of recordings. The software outputs not only graphical representations of the data, as shown in Fig 9, but also a quantitative summary (in a separate text box below the plot) of distribution parameters and log-likelihood ratio (LLR) comparisons between each heavy-tailed distribution considered.

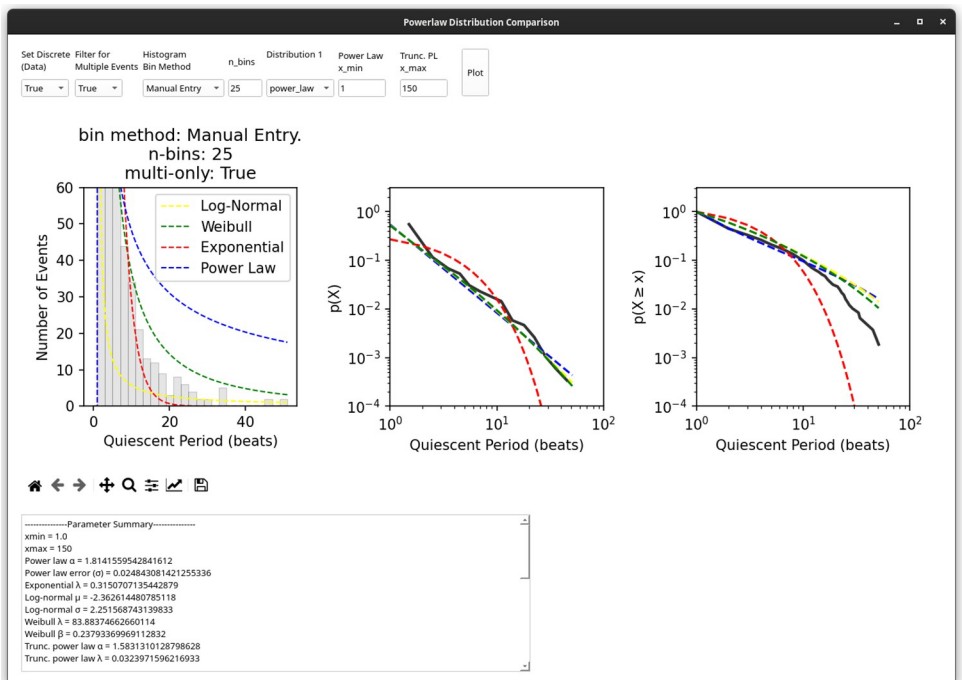

**Fig 9. 'Powerlaw Distribution Comparison' window.** The results of power law analysis, both qualitative (plots) and quantitative (numerical output, text box below plots) are presented in this window of the application. The user can adjust a variety of parameters, as necessary, and execute their analysis by clicking 'Plot'. The power law plot was adapted from [25].

Using Cardio PyMEA's power law analysis integration and pacemaker translocation detection tool revealed that, while pacemaker translocations do not necessarily follow a general power law distribution, they do convincingly obey a doubly truncated power law distribution (i.e. a power law with an exponentially-defined maximum, or cutoff, value that captures finite-size effects within the system) with a power law exponent of = -1.583 [25]. Several known power law-obeying systems demonstrate a similar exponent of with = -1.5 and many are categorized as critical systems. Some suspected critical systems, e.g. neuronal cultures, exhibit optimal performance (e.g. maximized information exchange between individual units within the system) when the system is operating at or near a critical point [45–47]. The value for calculated from pacemaker translocations suggests that critical dynamics may play a role in maximizing information transfer between cardiomyocytes within the culture and facilitate the identification of an optimal pacemaker region of the culture [25]. Thus, Cardio PyMEA facilitates and greatly simplifies power law analysis of cardiomyocyte cultures in order to help elucidate potential mechanisms governing cardiomyocyte system dynamics. Furthermore, thanks to Cardio PyMEA's extensible design philosophy, it would be quite feasible to further adapt the software for power law analysis of other observable phenomena in cardiomyocyte networks. Such work could constitute a future development goal.

## Cardio PyMEA helps democratize computational tools for cardiomyocyte analysis

A key aim underlying Cardio PyMEA's development was to provide both a free-to-use and free-to-edit computational tool for cardiomyocyte MEA analysis. Most, if not all, previously released, free-to-use cardiomyocyte MEA analysis applications are written in MATLAB.

Although these programs can be distributed freely, any efforts to edit and execute the underlying source code require a MATLAB license. This presents a potentially high barrier-to-entry in the form of operational costs, particularly at the academic tier for smaller institutions, that can be burdensome for researchers [24]. Thus, it became clear that achieving the elimination of these barriers to accessibility and development would require writing Cardio PyMEA in Python.

Python is an established, mature, and robust programming language known for its accessibility to those new to computer programming and its ubiquity in data science and machine learning applications. It provides a minimal barrier to entry, requiring only a computer and an internet connection. Python enjoys widespread community support via online resources which provide abundant resources for fledgling and senior programmers alike [48].

Cardio PyMEA leverages the aforementioned characteristics of Python to provide a cardiomyocyte MEA analysis tool suitable for continued development and refinement. The software's GUI was written using PyQt5 and adheres to a basic object-oriented programming paradigm that, with a modest background, is simple enough for a novice programmer to grasp and modify. In fact, Cardio PyMEA was designed in such a way that undergraduate students with no or modest programming experience could (and, in fact, did) successfully contribute new functionalities to the software. Therefore, Cardio PyMEA should prove suitable as an ongoing developmental platform and help to further democratize analytical tools for cardiomyocyte MEA data analysis.

## Conclusion and outlook

Cardio PyMEA is an open-source software designed to facilitate the analysis of microelectrode array data obtained for cardiomyocyte cultures. It provides an easy-to-use graphical user interface to allow non-programmers to carry out analysis of MEA data and, because of its object oriented nature, can be readily modified to accommodate additional MEA geometries and configurations. Calculations for several cardiomyocyte properties were successfully executed using Cardio PyMEA, including time lag, local activation time, upstroke and conduction velocities, beat amplitude, and beat interval, among others. Cardio PyMEA is readily extensible and provides users with two built-in analytical techniques for the assessment of property-distance correlations and the observance of power laws in pacemaker translocations.

Choosing to write Cardio PyMEA in Python presented clear benefits in the form of open access, extensibility, and ease of use. However, there are some drawbacks inherent to this choice that merit a brief discussion. Among the more significant issues inherent to Python application development is the global interpreter lock (GIL). When executing a Python script or program, the interpreter occupies one–and only one–computational thread on the host machine's CPU. This causes the GUI's widgets (e.g. the file dialog box) to temporarily "freeze" during the execution of a function call, such as a calculation or plotting operation, until the operation has completed. As a consequence of this GIL, the user may experience some minor performance issues during the operation of the program in the form of windows freezing, accompanied by a request to wait for the program to complete its task. One way this issue could be remedied is through extensive revisions to the code to enact multithreaded operations–a non-trivial exercise, but one which may take place at a later date.

Another opportunity for improvement, in select cases, is input/output and calculation speed. Python is known to be a relatively slow programming language, particularly compared to C and C++. Loading files is one of the more time-intensive operations performed by Cardio PyMEA, particularly for *.txt files exceeding 500 megabytes in size. For example, on Linux kernel 5.16, the import operation for a file of size 1.9GB (e.g. a 20 minute long, 120 electrode

MEA recording) can take ~45 seconds. Although our development was limited to *.txt file inputs as a result of MC_Data file outputs, it may be possible to incorporate a conversion tool into Cardio PyMEA that yields hierarchical data format HDF5 files for subsequent loading operations. Use of HDF5 input files has the potential to further improve data load times and partially alleviate data storage requirements [49].

Additionally, improvements to graphing performance may be possible by exchanging function calls to the Matplotlib and Seaborn plotting libraries for equivalents in PyQtGraph. PyQt-Graph can demonstrate superior speed compared to the aforementioned libraries [50]. Many calculations in Cardio PyMEA were optimized through the use of libraries like NumPy and Numba, which utilize C-based operations, to perform similarly to analogous calculations in other, faster programming languages. Further optimization should be possible with additional development.

Beyond performance improvements, we expect to see Cardio PyMEA acquire new features and functionalities over time. One area that may be primed for expanded development is comparative analysis of MEAs. Implementation of comparative methods, in which two separate recording files are loaded and analyzed in parallel, may provide significant workflow improvements over the current paradigm. Ultimately, Cardio PyMEA's potential is bound only by the imagination of its contributors and evolution of the Python programming language.

## Supporting information

**S1 File. Tutorial.** User tutorial in PDF (*.pdf) format for adding new MEA geometries to Cardio PyMEA.
(DOCX)

**S2 File. Save file.** Example save file in spreadsheet (*.xlsx) format.
(XLSX)

**S3 File. Batch file.** Example batch file in spreadsheet (*.xlsx) format.
(XLSX)

## Acknowledgments

CSD would like to acknowledge the work of Michelle (Nguyen) Phi, at the time an undergraduate biochemistry student at UCLA, for providing invaluable assistance to MEA software development during the earliest stages of this project. Her contributions to the original codebase were essential in advancing this project.

## Author Contributions

**Conceptualization:** Christopher S. Dunham, Atsushi Nakano, Adam Z. Stieg, James K. Gimzewski.

**Data curation:** Christopher S. Dunham, Haruko Nakano.

**Formal analysis:** Christopher S. Dunham, Madelynn E. Mackenzie.

**Funding acquisition:** Atsushi Nakano, Adam Z. Stieg, James K. Gimzewski.

**Investigation:** Christopher S. Dunham, Adam Z. Stieg, James K. Gimzewski.

**Methodology:** Christopher S. Dunham, Haruko Nakano, Atsushi Nakano, Adam Z. Stieg.

**Project administration:** Christopher S. Dunham, Adam Z. Stieg, James K. Gimzewski.

**Resources:** Haruko Nakano, Atsushi Nakano, Adam Z. Stieg, James K. Gimzewski.

**Software:** Christopher S. Dunham, Madelynn E. Mackenzie.

**Supervision:** Atsushi Nakano, Adam Z. Stieg, James K. Gimzewski.

**Validation:** Christopher S. Dunham, Madelynn E. Mackenzie, Alexis R. Kim, Atsushi Nakano, James K. Gimzewski.

**Visualization:** Christopher S. Dunham.

**Writing – original draft:** Christopher S. Dunham, Madelynn E. Mackenzie.

**Writing – review & editing:** Christopher S. Dunham, Madelynn E. Mackenzie, Haruko Nakano, Alexis R. Kim, Atsushi Nakano, Adam Z. Stieg, James K. Gimzewski.

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
