## [Decision Letter · Decision Letter 0]

13 Apr 2022

PONE-D-22-08656Cardio PyMEA: A user-friendly, open-source Python application for cardiomyocyte microelectrode array analysisPLOS ONE

Dear Dr. Dunham,

Thank you for submitting your manuscript to PLOS ONE. After careful consideration, we feel that it has merit but does not fully meet PLOS ONE’s publication criteria as it currently stands. Therefore, we invite you to submit a revised version of the manuscript that addresses the points raised during the review process. Both reviewers indicated that the paper needs minor improvements in order to be accepted by PLOS ONE. Read the comments carefully and address the issues raised by the reviewers.  

We look forward to receiving your revised manuscript.

Kind regards,

Rafael Sachetto Oliveira, Ph.D

Academic Editor

PLOS ONE

Journal Requirements:

Reviewers' comments:

Reviewer's Responses to Questions

**Comments to the Author**

1. Is the manuscript technically sound, and do the data support the conclusions?

Reviewer #1: Yes

Reviewer #2: Yes

2. Has the statistical analysis been performed appropriately and rigorously? 

Reviewer #1: Yes

Reviewer #2: N/A

3. Have the authors made all data underlying the findings in their manuscript fully available?

Reviewer #1: No

Reviewer #2: Yes

4. Is the manuscript presented in an intelligible fashion and written in standard English?

Reviewer #1: Yes

Reviewer #2: Yes

5. Review Comments to the Author

Reviewer #1: This article presents Cardio PyMEA, a new Python-based tool for cardiomyocyte microelectrode array (MEA) postprocessing with advanced statistical analysis functionalities. The software offers a cross-platform, open-source alternative to previous similar options, typically implemented for Matlab, a proprietary software. The authors stress that Cardio PyMEA is extensible to a great degree, easily adaptable to different experimental configurations.

The software functions have been tested and validated using 30 measurements on 3 different cardiomyocyte differentiations. Cardio PyMEA is already available, but the data collection used in the article is not.

The article is well-written, accessible to the audience, and enjoyable to read. However, certain aspects would benefit from an improvement to increase the general quality of the article.

0. The lack of line numbers represents a difficulty when referencing the manuscript.

1. Scientific novelty is questionable in many aspects, which imposes the need of highlighting the unique features offered by Cardio PyMEA. The use of open-source software instead of Matlab is a great feature but does not justify the manuscript by itself.

a) In general, the paper should focus more on the advantages or quality of this tool vs. others.

b) Section title using the word "unique" indicates that other sections may not be that unique. This section is crucial. Consider the expansion of this section and a reference to figures, not only for showing results, but also to complement the description of the user interface.

2) What if the MEA equipment used is from a different manufacturer than Multichannel Systems? Would the simplest option be to replicate the input format described in the text and create a suitable ElectrodeConfig?

3) Performance with a noisy signal is especially relevant. Therefore, the sentence below would benefit from referring to results or examples, instead of just a statement that the reader must believe.

"Cardio PyMEA demonstrated effective performance under noisy conditions (Figs 6E-F) thanks to its permission of user-provided parameter inputs. User-defined adjustments to beat amplitude and distance thresholds provide flexibility in analyzing noisy datasets that could otherwise become intractable under automated beat detection algorithms."

4) The sentence below is confusing, please consider representing F7 in Figure 3.

"The second is expected to correspond to the electrode F7. This electrode is designated as channel 1 (electrode 1) in the manufacturer’s MEA channel schematic."

5) The relevance of power-law analysis in this context should be described in Methods, and at least mentioned in the Introduction. Currently, the Methods section describes the power-law analysis briefly and in a very abstract manner. Why this is relevant here remains unclear. Power law application is explained in Results and Discussion, but further effort is required to inform the reader about any conclusions she/he may or may not obtain from Figure 9.

Given that the power-law analysis is a unique point of this software, a more instructive approach should be taken to allow the reader to understand how she/he can benefit from this functionality.

Thanks for an enjoyable read.

Best wishes.

Reviewer #2: The present work presents an open-source Python graphical application for the study of microelectrode array analysis (MEA) of cardiomyocytes. The application offers a series of fundamental features for loading, analyzing, and exporting the data from MEA recordings. A series of calculations such as beat detection, electrode exclusion, electrode silencing, pacemaker origin detection, local activation time, maximum upstroke velocity, beat amplitude and interval, conduction velocity, pacemaker translocation detection, and statistical analyses are available. The manuscript is very well written, well organized, relevant, and the presented results are encouraging and significant. The reviewer has no major or general comments, since the manuscript and the contribution are already very clear and significant. However a list of suggestions for improving the text of the manuscript and for future upgrades on the software are provided below, together with some specific or minor comments.

Suggestions for improving the manuscript

1. Sample data for immediate usage and test of PyMEA could be made available in the Github repository.

2. The abstract is very focused on the software license and other computational aspects of PyMEA. However, the reviewer believes that providing a compact and general overview of the main capabilities of the software (beat detection, pacemaker origin estimation, and

etc) on the abstract could further gain the attention of researchers working within this field and looking for this kind of solution.

3. The authors could explain the method used to compute the field potential duration (FPD) according to the mentioned reference using a simple and compact approach. All the other details from the features of the software are well explained, with the exception of this

calculation.

Specific comments and minor suggestions

1. No page or line numbering are provided in the manuscript. This makes the process of reviewing a little bit harder.

2. Section "Automated electrode exclusion". Instead of using the term data modification, the reviewer thinks that data treatment or data pre-processing is more adequate.

3. After the equation of the radius r, it is said: These parameters are additionally bound. It seems the authors are referring to imposing bounds on the parameters during the least-squares fitting. Please make it clearer.

4. LAT was defined 2 times or more.

5. Please specify what type of finite difference was applied to approximate the derivatives for the maximum upstroke velocity calculation (forward,backward,central,...).

6. Conduction velocity calculation. Although very simple and easy to infer, the formula "electrode-distance/∆LAT ", lacks the definition of ∆LAT .

7. Property vs distance from pacemaker. Please define multiplier

8. In general the texts describing the name of functions within the code are written with another font style. For LaTeX users this is usually done with the command texttt.

9. Please explain or provide a reference for Sturge’s rule.

10. It seems the concepts of power-law and probability density functions are mixed or entwined. The authors could try to improve and distinguish this a little bit.

11. Should some signal processing tools be used for noisy signals before MEA calculations? Is this a common practice within the field? Perhaps the authors could mention something about this.

12. Cardio PyMEA provides uniques (section). Before the caption of Fig.8 the authors mention "suggesting that there is a strong relationship between these two properties", when refering to the R 2 coefficient of determination. Perhaps using "suggesting a good fitting was achieved", or something similar would be better, to avoid a possible confusion with a positive correlation.

Suggestions for future improvements of the software

1. At the final section of the manuscript the authors mention the software may freeze when using large datasets. On the reviewer’s experience this may be related to the usage of the matplotlib as a backend for the plots. Although it is pretty standard forPython code to use matplotlib, for huge datasets it takes significant time to render and plot data with this library. An alternative is to use the PyQtGraph library

(https://www.pyqtgraph.org/), which is faster and can easily handle large datasets. It can also deliver high-quality plots.

2. Currently PyMEA only supports input data in the format of a specific MEA recording software. For a broader use, this could be extended to other input formats which might be also in use within the MEA community.

3. Also with respect to input/output, the authors could consider in the future using data in the HDF5 file format. It can significantly reduce storage of large datasets, and may provide a faster interface for reading and writing data, which could also impact on the performance

of the software when loading large datasets.

4. Please check the OpenCARP suite for cardiac electrophysiology, which provides the limpetgui, a tool for visualization of large datasets of cardiac action potential models. It uses both PyQtGraph and HDF5. Link: https://opencarp.org/documentation/examples/visualization/limpetgui

5. Another interesting feature for the future, would be the possibility of loading two datasets for comparisons and computation of quantitive metrics about their differences, in case this could be useful for MEA studies.

6. PLOS authors have the option to publish the peer review history of their article (what does this mean?). If published, this will include your full peer review and any attached files.

Reviewer #1: No

Reviewer #2: No

---

## [Author Response · Author response to Decision Letter 0]

7 May 2022

The authors would like to thank the reviewers and the editorial staff for their commentary and recommendations regarding this manuscript. We are grateful that this manuscript was received positively. We have carefully reviewed the editorial and reviewer responses and agree with many of the recommended modifications. We have since refined the manuscript in accordance with the editorial and reviewer suggestions and believe it has been greatly enhanced relative to the original submission. 

Please see below for our responses to each comment. Thank you again for your consideration and generous feedback on our manuscript. Additionally, please refer to the attached Response to Reviewers document for easier reading (responses are color-coded in blue text in the attached document).

Editor:

and 

Response: We apologize for the omission of line and page numbers, and have since corrected the manuscript submission to include them. We believe the manuscript now adheres to all stated formatting requirements.

Response: Thank you for this comment regarding code sharing practices. In addition to the Github repository provided in the manuscript (https://github.com/csdunhamUC/cardio_pymea), a full snapshot of the code (*.py files) was uploaded to Data Dryad/Zenodo at the following DOI: https://doi.org/10.5281/zenodo.6462799. 

The data submitted to Data Dryad will be available at the following DOI: https://doi.org/10.5068/D14H5C. These code and data submissions, respectively, remain under review at the time of this response, but should complete curation and go live in the coming days.

Response: No references were removed from the manuscript. Several references were added to the manuscript in light of reviewer feedback. 

The following references were added to the manuscript:

Bigger JT, Steinman RC, Rolnitzky LM, Fleiss JL, Albrecht P, Cohen RJ. Powerlaw behavior of RR-interval variability in healthy middle-aged persons, patients with recent acute myocardial infarction, and patients with heart transplants. Circulation. 1996;93: 2142–2151. pmid:8925583

Scott DW. Sturges' rule. Wiley Interdiscip Rev Comput Stat. 2009;1: 303-306. doi: 10.1002/wics.35

Beggs J, Plenz D. Neuronal Avalanches in Neocortical Circuits. J Neurosci. 2003;23: 11167–11177. pmid:14657176

Dunham CS, Lilak S, Hochstetter J, Loeffler A, Zhu R, Chase C, et al. 2021. Nanoscale neuromorphic networks and criticality: a perspective. J Phys Complex. 2021;2: 042001.

Meijers M, Ito S, Rein ten Wolde P. Behavior of information flow near criticality. Phys Rev E. 2021;103: L010102. pmid:33601642

Collette A. Python and HDF5 [Internet]. Sebastopol (CA): O'Reilly Media, Inc (US); [reviewed 2022 Apr 17; cited 2022 Apr 18]. Available from: https://www.oreilly.com/library/view/python-and-hdf5/9781491944981/ch01.html

PyQtGraph [Internet]. PyQtGraph; [reviewed 2022 Apr 17; cited 2022 Apr 18]. Available from: https://www.pyqtgraph.org

We confirm that the reference list is complete and accurate.

Reviewer #1:

This article presents Cardio PyMEA, a new Python-based tool for cardiomyocyte microelectrode array (MEA) postprocessing with advanced statistical analysis functionalities. The software offers a cross-platform, open-source alternative to previous similar options, typically implemented for Matlab, a proprietary software. The authors stress that Cardio PyMEA is extensible to a great degree, easily adaptable to different experimental configurations.

The software functions have been tested and validated using 30 measurements on 3 different cardiomyocyte differentiations. Cardio PyMEA is already available, but the data collection used in the article is not.

The article is well-written, accessible to the audience, and enjoyable to read. However, certain aspects would benefit from an improvement to increase the general quality of the article.

0. The lack of line numbers represents a difficulty when referencing the manuscript.

Response: We apologize for this omission and have fixed the issue for the revision.

1. Scientific novelty is questionable in many aspects, which imposes the need of highlighting the unique features offered by Cardio PyMEA. The use of open-source software instead of Matlab is a great feature but does not justify the manuscript by itself.

a) In general, the paper should focus more on the advantages or quality of this tool vs. others.

Response: Thank you for this suggestion. We’ve added language throughout the manuscript to better emphasize the advantages of this tool against others and unique properties. These include expansion and highlighting of power law analysis workflow and property-distance relationships, as well as extensibility of the software relative to commercial solutions

b) Section title using the word "unique" indicates that other sections may not be that unique. This section is crucial. Consider the expansion of this section and a reference to figures, not only for showing results, but also to complement the description of the user interface.

Response: We significantly expanded content related to the power law analysis features in Cardio PyMEA and added language to further emphasize the features that are unique to this software.

2) What if the MEA equipment used is from a different manufacturer than Multichannel Systems? Would the simplest option be to replicate the input format described in the text and create a suitable ElectrodeConfig?

Response: Thank you for this question. You are correct, that would be the simplest option. We are also interested in adding data acquired from multi-well plate-based systems and welcome the sharing of data that would facilitate its implementation.

3) Performance with a noisy signal is especially relevant. Therefore, the sentence below would benefit from referring to results or examples, instead of just a statement that the reader must believe.

"Cardio PyMEA demonstrated effective performance under noisy conditions (Figs 6E-F) thanks to its permission of user-provided parameter inputs. User-defined adjustments to beat amplitude and distance thresholds provide flexibility in analyzing noisy datasets that could otherwise become intractable under automated beat detection algorithms."

Response: Thank you for this comment. We have rephrased the sentences in question to highlight user-defined inputs as they relate to the detection of beats under low signal-to-noise conditions. We also struck the language making a different comparison to automated beat detection algorithms. Finally, we added approximate S/N ratios to Fig 6D-F to better illustrate the comparison. The new wording is as follows:

“Cardio PyMEA demonstrated effective performance under noisy conditions thanks to its utilization of user-provided parameter inputs. User-defined thresholds of beat amplitude and distance provide flexibility in analyzing noisy datasets, even under low signal-to-noise (S/N) conditions (Figs 6E-F).”

4) The sentence below is confusing, please consider representing F7 in Figure 3.

"The second is expected to correspond to the electrode F7. This electrode is designated as channel 1 (electrode 1) in the manufacturer’s MEA channel schematic."

Response: Thank you for raising this concern. Figure 3 has since been updated to include all the electrode designations for ease of understanding. 

5) The relevance of power-law analysis in this context should be described in Methods, and at least mentioned in the Introduction. Currently, the Methods section describes the power-law analysis briefly and in a very abstract manner. Why this is relevant here remains unclear. Power law application is explained in Results and Discussion, but further effort is required to inform the reader about any conclusions she/he may or may not obtain from Figure 9.

Given that the power-law analysis is a unique point of this software, a more instructive approach should be taken to allow the reader to understand how she/he can benefit from this functionality.

Response: Thank you for these suggestions. We’ve added a brief line to the Introduction that highlights one area of importance concerning power law analysis and moved a citation to better orient the reader to this concept. We moved select content from the Results and Discussion, and added further context, to help the reader understand the importance of power laws as they relate to cardiomyocyte cultures to the Methods section. Finally, we revised the Results and Discussion sub-section concerning power law analysis to further elaborate on both the new Methods content and the benefits of the analysis. We believe this area is significantly improved over the previous iteration.

Reviewer #2:

The present work presents an open-source Python graphical application for the study of microelectrode array analysis (MEA) of cardiomyocytes. The application offers a series of fundamental features for loading, analyzing, and exporting the data from MEA recordings. A series of calculations such as beat detection, electrode exclusion, electrode silencing, pacemaker origin detection, local activation time, maximum upstroke velocity, beat amplitude and interval, conduction velocity, pacemaker translocation detection, and statistical analyses are available. The manuscript is very well written, well organized, relevant, and the presented results are encouraging and significant. The reviewer has no major or general comments, since the manuscript and the contribution are already very clear and significant. However a list of suggestions for improving the text of the manuscript and for future upgrades on the software are provided below, together with some specific or minor comments.

Suggestions for improving the manuscript

1. Sample data for immediate usage and test of PyMEA could be made available in the Github repository.

Response: Thank you for this suggestion. We have formatted and uploaded to Github a small sample file for this purpose. The file name is “Dummy Data 12 Beats.txt”. Additionally, the data used in this manuscript will be made available via Data Dryad at the following DOI: https://doi.org/10.5068/D14H5C. This URL will also be included in the Github README once it is active.

2. The abstract is very focused on the software license and other computational aspects of PyMEA. However, the reviewer believes that providing a compact and general overview of the main capabilities of the software (beat detection, pacemaker origin estimation, and

etc) on the abstract could further gain the attention of researchers working within this field and looking for this kind of solution.

Response: Thank you for this comment. We have amended the abstract to better highlight the primary functions and uses of the Cardio PyMEA system with respect to cardiomyocyte culture analysis. 

3. The authors could explain the method used to compute the field potential duration (FPD) according to the mentioned reference using a simple and compact approach. All the other details from the features of the software are well explained, with the exception of this

calculation.

Response: Thank you for this recommendation. We have since expanded upon the paragraph for the FPD calculation to provide a brief summary of the methods from the reference. The new sentences are as follows and begin immediately following the mention of Vázquez-Seisdedos et al.:

“A brief summary of this method follows. First, the T-wave is identified using a peak finding algorithm akin to the R-wave. Second, the maximum first derivative of the field potential signal between the T-wave peak and baseline is identified. Using the location (i.e. time or x-value) of the maximum derivative as a fixed point, along with an arbitrary location 50-200 ms past the T-wave peak, the algorithm fits a trapezoid with a mobile point to the two fixed points. Optimizing the maximum area of the trapezoid yields the T-wave endpoint”

Specific comments and minor suggestions

1. No page or line numbering are provided in the manuscript. This makes the process of reviewing a little bit harder.

Response: We apologize for this omission and have fixed the issue for the revision.

2. Section "Automated electrode exclusion". Instead of using the term data modification, the reviewer thinks that data treatment or data pre-processing is more adequate.

Response: This is a great suggestion and we have updated the phrasing to “data pre-processing”.

3. After the equation of the radius r, it is said: These parameters are additionally bound. It seems the authors are referring to imposing bounds on the parameters during the least-squares fitting. Please make it clearer.

Response: Thank you for noting this. We have improved upon this sentence by changing it to the following: 

“The radius parameter, r, is bound within the geometry of the MEA during the fitting process. This measure is taken to ensure that the center of the circle, (h, k), does not reside outside of the culture region of the MEA.”

4. LAT was defined 2 times or more.

Response: Thank you for spotting this repetition. We have removed the redundant definition.

5. Please specify what type of finite difference was applied to approximate the derivatives for the maximum upstroke velocity calculation (forward,backward,central,...).

Response: Thank you for this suggestion. We have revised the sentence in question to clearly state that we are using a backward finite difference method.

6. Conduction velocity calculation. Although very simple and easy to infer, the formula "electrode-distance/∆LAT ", lacks the definition of ∆LAT .

Response: Thank you for noting this. We have added the definition for ∆LAT in the sentence in question, just prior to stating the formula. The sentence now reads: 

“Conduction velocity is calculated in a straightforward manner by determining the distance between any two electrodes and dividing this distance by the change in local activation time (ΔLAT) recorded at the electrodes, i.e.: (electrode distance)/ΔLAT.”

7. Property vs distance from pacemaker. Please define multiplier

Response: Thank you for raising this concern. We agree this term was confusing and changed the phrasing of the sentence to make it clear that we are referring to the number of standard deviations. The sentence now reads as:

“Any values outside of the range specified by the (number of standard deviations * Sigma) operation are excluded from the analysis.”

8. In general the texts describing the name of functions within the code are written with another font style. For LaTeX users this is usually done with the command texttt.

Response: Thank you for highlighting this detail. All descriptions of function names from within the code have since been changed to the font Courier New.

9. Please explain or provide a reference for Sturge’s rule.

Response: Thank you for this suggestion. We’ve added a reference for Sturges’ rule and fixed the misspelling. The reference is: Scott DW. Sturges' rule. Wiley Interdiscip Rev Comput Stat. 2009;1: 303-306. doi: 10.1002/wics.35

10. It seems the concepts of power-law and probability density functions are mixed or entwined. The authors could try to improve and distinguish this a little bit.

Response: Thank you for this suggestion. We have improved the language in the section “Cardio PyMEA simplifies the analysis of pacemaker stability for power law behavior” to clearly identify power laws as probability distributions and necessitate appropriate considerations during their analysis. This section was also expanded upon following comments from Reviewer 1. We believe the section now does a better job of not only communicating what power laws are, but also why we care about them in regard to cardiac systems.

11. Should some signal processing tools be used for noisy signals before MEA calculations? Is this a common practice within the field? Perhaps the authors could mention something about this.

Response: Thank you for this question. Signal processing is often done within the field during MEA analysis, depending on how noisy the original recording/signals are. The analysis presented in this work did not utilize any signal processing steps as many of the MEA recordings presented fairly clean signals overall. 

However, Cardio PyMEA does offer the option to perform signal processing during the beat detection stage. As noted in the “Beat Detection” sub-section of the Methods section, users can choose from low-pass, high-pass, and bandpass filters. The underlying function calls made by Cardio PyMEA rely on signal processing tools from the SciPy library. 

We amended a short phrase to an early sentence in the results and discussion that emphasizes that the results presented within were not subject to signal filters.

“The beat detection algorithm was successful in identifying beats from the field potential signals without the need for signal filtering tools”

12. Cardio PyMEA provides uniques (section). Before the caption of Fig.8 the authors mention "suggesting that there is a strong relationship between these two properties", when refering to the R 2 coefficient of determination. Perhaps using "suggesting a good fitting was achieved", or something similar would be better, to avoid a possible confusion with a positive correlation.

Response: Thank you for this recommendation. We have adjusted the language to read: “suggesting that a relationship exists, based on good fitting between these two properties, for the given cardiomyocyte network.”

Suggestions for future improvements of the software

1. At the final section of the manuscript the authors mention the software may freeze when using large datasets. On the reviewer’s experience this may be related to the usage of the matplotlib as a backend for the plots. Although it is pretty standard forPython code to use matplotlib, for huge datasets it takes significant time to render and plot data with this library. An alternative is to use the PyQtGraph library

(https://www.pyqtgraph.org/), which is faster and can easily handle large datasets. It can also deliver high-quality plots.

Response: This is a very welcome and helpful suggestion. We agree that some performance improvements could be obtained by migrating away from Matplotlib to PyQtGraph. Under typical analysis workflows, the most common performance issue we see is in file loading, which can take some time for large files (exceeding e.g. 500mb; most of our files were in the ~2gb range). We agree that your other suggestions about different file types would almost certainly help address this performance bottleneck, although there are other factors affecting that particular area (discussed below). Nevertheless, this is certainly a welcome development suggestion that we will consider pursuing. We added a note regarding Matplotlib -> PyQtGraph migration to a new paragraph in the Outlook section.

2. Currently PyMEA only supports input data in the format of a specific MEA recording software. For a broader use, this could be extended to other input formats which might be also in use within the MEA community.

Response: We agree whole-heartedly with this sentiment. It is our hope that groups with access to other types of MEA systems will collaborate with us to implement other formats, e.g. multi-well plate systems.

3. Also with respect to input/output, the authors could consider in the future using data in the HDF5 file format. It can significantly reduce storage of large datasets, and may provide a faster interface for reading and writing data, which could also impact on the performance of the software when loading large datasets.

Response: We are highly amenable to this suggestion. The primary reason for utilizing CSV/text files at this stage is due to the output of the MC Data conversion tool from Multichannel Systems. This tool most conveniently produces ASCII files (in the form of CSV/text files) and does not have support for file types such as HDF5. Implementing a converter tool into Cardio PyMEA to generate HDF5 files is a great suggestion for future development. We have included a paragraph about these suggestions in the Outlook section.

4. Please check the OpenCARP suite for cardiac electrophysiology, which provides the limpetgui, a tool for visualization of large datasets of cardiac action potential models. It uses both PyQtGraph and HDF5. Link: https://opencarp.org/documentation/examples/visualization/limpetgui

Response: This is great! Thanks for informing us about this fantastic simulation software. We will dig into this further and see what inspiration we can draw for future development.

5. Another interesting feature for the future, would be the possibility of loading two datasets for comparisons and computation of quantitive metrics about their differences, in case this could be useful for MEA studies.

Response: This is a great idea! This is something that we will absolutely keep in mind for ongoing development and we added commentary about this in the form of a short paragraph at the end of the Outlook section. We whole-heartedly welcome collaboration with any groups interested in contributing toward this implementation.

---

## [Decision Letter · Decision Letter 1]

17 May 2022

Cardio PyMEA: A user-friendly, open-source Python application for cardiomyocyte microelectrode array analysis

PONE-D-22-08656R1

Dear Dr. Dunham,

We’re pleased to inform you that your manuscript has been judged scientifically suitable for publication and will be formally accepted for publication once it meets all outstanding technical requirements.

Kind regards,

Rafael Sachetto Oliveira, Ph.D

Academic Editor

PLOS ONE

Additional Editor Comments (optional):

Reviewers' comments:

Reviewer's Responses to Questions

**Comments to the Author**

1. If the authors have adequately addressed your comments raised in a previous round of review and you feel that this manuscript is now acceptable for publication, you may indicate that here to bypass the “Comments to the Author” section, enter your conflict of interest statement in the “Confidential to Editor” section, and submit your "Accept" recommendation.

Reviewer #1: All comments have been addressed

Reviewer #2: All comments have been addressed

2. Is the manuscript technically sound, and do the data support the conclusions?

Reviewer #1: Yes

Reviewer #2: Yes

3. Has the statistical analysis been performed appropriately and rigorously? 

Reviewer #1: N/A

Reviewer #2: N/A

4. Have the authors made all data underlying the findings in their manuscript fully available?

Reviewer #1: Yes

Reviewer #2: Yes

5. Is the manuscript presented in an intelligible fashion and written in standard English?

Reviewer #1: Yes

Reviewer #2: Yes

6. Review Comments to the Author

Reviewer #1: Thanks for the extensive work conducted to incorporate the feedback offered by the editor and reviewers.

One last issue for me was the availability of a data sample to test the software, but I see that this has been solved too.

I hope that you consider that this revision process has strengthened your manuscript.

Best wishes.

Reviewer #2: All the issues and suggestions raised by the reviewer were fully addressed.

This improved the quality and readability of the manuscript.

7. PLOS authors have the option to publish the peer review history of their article (what does this mean?). If published, this will include your full peer review and any attached files.

Reviewer #1: **Yes: **Hector Martinez-Navarro

Reviewer #2: **Yes: **Bernardo Martins Rocha, Universidade Federal de Juiz de Fora

---

## [Editor Report · Acceptance letter]

18 May 2022

PONE-D-22-08656R1 

Cardio PyMEA: A user-friendly, open-source Python application for cardiomyocyte microelectrode array analysis 

Dear Dr. Dunham:

I'm pleased to inform you that your manuscript has been deemed suitable for publication in PLOS ONE. Congratulations! Your manuscript is now with our production department. 

Kind regards, 

on behalf of

Dr. Rafael Sachetto Oliveira 

Academic Editor

PLOS ONE